# Utilisation of tools to facilitate cross-border communication during international food safety events, 1995–2019: a realist synthesis protocol

Carmen Joseph Savelli [ID] ,[1,2] Ceu Mateus[2]

[1]Food Safety and Zoonoses, World Health Organization, Geneva, Switzerland
[2]Division of Health Research, Faculty of Health and Medicine, Lancaster University, Lancaster, United Kingdom

**Correspondence to**
Carmen Joseph Savelli;
savellic@who.int

## ABSTRACT

**Introduction** Efficient communication and coordination between countries is needed for prevention, detection and response to international food safety events. While communication tools exist, current evidence suggests that they are only effective within certain contexts and only cover certain geographic areas. There is a need to unpack and explore the mechanisms of how and in what context such communication tools and their components are effective at facilitating international communication and coordination to keep food safe and mitigate the burden of foodborne disease around the globe.

**Methods and analysis** A realist synthesis will be undertaken to understand how and why certain processes and structures of communication tools, used during international food safety events, influence their utility and effectiveness according to different contextual factors. The focus of this review is explanatory and aims to develop and refine theory regarding how contextual factors trigger specific processes and mechanisms to produce outcomes. Using the realist context–mechanism–outcome configuration of theory development, a range of sources have been used to develop the initial programme theory, including the author's experience, a scoping review of published papers and grey literature and input from an expert reference committee. To support, expand or refute the initial theory, data will be synthesised from published literature and input from the expert reference committee.

**Ethics and dissemination** Ethical approval is not required for this review as it does not involve primary research. However, it will be conducted according to the appropriate ethical standards of accuracy, utility, usefulness, accountability, feasibility and propriety. The RAMESES publication standards will be followed to report the findings of this review. On completion, the final manuscript will be shared with members of the FAO/WHO International Food Safety Authorities Network (INFOSAN) and published in a peer-reviewed journal.

## BACKGROUND

Access to sufficient amounts of safe and nutritious food is a basic requirement for human health. However, around the world unsafe food is known to cause more than 200 acute and chronic diseases, ranging from diarrhoea to cancer.[1] In 2015, the first estimates

of the global burden of foodborne diseases were reported by the WHO, indicating that 31 hazards (including bacteria, viruses, parasites, toxins and chemicals) were responsible for 600 million cases of foodborne diseases and 420 000 deaths worldwide in 2010.[2] This burden was disproportionately felt by children under 5 years of age who accounted for 40% of foodborne disease cases and 125 000 deaths.[2] While foodborne diseases are observed worldwide, Africa, South-East Asia and the Eastern Mediterranean regions report the highest burden.[2] In such high-burden areas, unsafe food presents additional consequences beyond disease burden, impeding socioeconomic development, overloading strained healthcare systems and damaging national economies, trade and tourism.[3] Furthermore, a 2018 study by the World Bank[4] indicates that unsafe food costs low-income and middle-income economies approximately US$100 billion in lost productivity and medical expenses each year.

**Strengths and limitations of this study**

► Uses realist methods to explore contextual factors and underpinning causal mechanisms of complex international communication during food safety events.
► Utilisation of an international expert review committee consisting of national government officials, international civil servants and academics to ensure complete coverage of the literature.
► Formulation of the context–mechanism–outcome programme theory relies heavily on published literature and therefore may be subject to publication bias.
► Review findings will be context-specific and therefore must be considered within the context of this research.

Foodborne diseases are preventable; however, prevention requires investment and coordinated action across multiple sectors to

strengthen national food safety systems. Multiple agencies responsible for health, agriculture, veterinary services, trade and several others must work together to build a strong and resilient national food safety system. The WHO has identified several core capacities that national governments should develop to safeguard national food supplies. The development of such core capacities is evaluated yearly by the WHO to determine whether countries have established functional mechanisms for the detection, prevention and response to foodborne disease and food contamination events. Data from 2017 indicate that 78% of the attributes of core capacities required for food safety have been developed globally, although disparities exist between regions. For example, 90% of the required core capacities have been achieved in Europe, while in Africa, only 54% of the core capacities have been achieved.[5]

An international food safety event results when unsafe food produced in one country is exported to at least one other country. Recent international food safety events have demonstrated that even in countries with well-developed capacities related to food safety, unsafe foods that are produced abroad and imported for domestic consumption have the potential to result in large-scale outbreaks of foodborne disease. For example, nearly 4000 people became infected with *Escherichia coli* (and nearly 800 developed haemolytic uremic syndrome) in Germany following the consumption of contaminated fenugreek sprouts, imported from Egypt in 2011. Illnesses related to the same imported product were concurrently reported in France.[6] In 2012, at least 11 000 cases of norovirus infection were reported in Germany following the consumption of frozen strawberries imported from China.[7] In 2008, 300 000 infants and children became ill in China, six of whom died, after consuming milk products contaminated with melamine. The contaminated products were directly exported or secondarily distributed to 47 countries around the world.[8] In 2013 and 2014, nearly 1500 cases of hepatitis A infection were identified in 13 European countries and linked to the consumption of internationally distributed frozen berries.[9] More recently, in 2017 and 2018 the world bore witness to the largest outbreak of listeriosis on record which occurred in South Africa and resulted in more than 1000 cases and 200 deaths. This protracted outbreak was eventually linked to domestically produced ready-to-eat meat products which were exported to 15 other countries in Africa.[10] Also in 2017 and 2018, an outbreak of salmonellosis in France affecting 37 infants was linked to contaminated infant formula that was exported worldwide to more than 80 countries.[11] These examples represent some of the largest international food safety events that have occurred in the recent past, either in terms of case-counts or number of countries affected, but smaller-scale events occur regularly. Furthermore, these events illustrate that even the most advanced food safety systems do not eliminate all foodborne hazards from reaching the public. The globalisation of our food supply means that unsafe food produced in one country can certainly result in cases of foodborne disease abroad.

Global food trade grew almost threefold from 2005 to 2015[12] and is projected to continue to rise.[13] Thus, there is a need for international coordination to facilitate rapid and efficient communication and collaboration between public health and food safety authorities (ie, competent authorities) worldwide to prevent, detect and respond to international food safety events when internationally traded food is deemed unsafe. Until relatively recently, timely mechanisms to facilitate such global communication did not exist. In the early 2000s, WHO Member States recognised this gap and adopted resolutions at the World Health Assemblies in 2000[14] and 2002[15] calling for improved communication and coordination during international food safety events, including better tools to facilitate this. Since then, advancements in communication technology have facilitated the development or expansion of international networks and knowledge sharing platforms to exchange molecular subtyping information of foodborne pathogens, epidemiologic information about foodborne diseases, as well as information on food contamination and related traceability details. Throughout this protocol, the term 'communication tool' will be used to encompass networks, knowledge sharing platforms, technical programmes or systems that facilitate communication related to food safety across national borders. These communication tools are complex for several reasons, because they represent disparate systems that may or may not interface with each other, operate in different languages, are coordinated by different institutions in different countries and are at various stages of development. Evidence from practice suggests that such tools are only effective within certain contexts and only cover certain geographic areas.[16–19] It is therefore necessary to unpack and explore the mechanisms of how and in what context such communication tools and their components are effective to facilitate international communication and coordination. Some examples of these communication tools include the European Rapid Alert System for Food and Feed (RASFF), the Association of Southeast Asian Nations (ASEAN) RASFF and the International Food Safety Authorities Network (INFOSAN). The European RASFF system is an example of a regional tool that works in the European context, in part because member countries adhere to the same legislation. The ASEAN RASFF system is an example of a tool that is less well established and member countries in this Asian context do not adhere to the same legislation. INFOSAN is a global tool, coordinated by the WHO and the Food and Agriculture Organization of the United Nations, but as described by Savelli et al,[20] a relatively limited number of active members from a select group of countries contribute most information exchanged through the network. Online supplementary file 1 provides a preliminary inventory of communication tools currently used or under development for exchanging information during international food safety events.

Unfortunately, a paucity of research has been conducted to investigate the attributes and effectiveness of the tools to facilitate cross-border communication during international food safety events. To date, most of the publications mentioning such tools focus on summarising a specific incident response, rather than explicitly examining the tools that were utilised. However, it is rather common for such reports of international food safety events to conclude by recommending that international efforts to strengthen rapid and efficient information exchange be improved through the further enhancement or utilisation of existing international networks and communication tools. These papers are typically written as outbreak reports rather than research studies.[8 21–29] Such reports also commonly refer to context-specific factors that facilitate or prevent rapid communication on various aspects of food safety investigations, such as poorly developed food safety systems, lack of national coordination, or limited technical capacity. Available research provides limited guidance for decision-makers coordinating international programmes that facilitate information exchange on food safety, on how to adopt best practices to achieve their objectives. In addition, as explained by Savelli et al,[20] the global food safety community would benefit from a thorough mapping of the interlinkages between such programmes and networks to better understand how they are being used, by whom and in what contexts. A realist synthesis is therefore proposed to begin to address this gap. The main question to guide this research is: how do different tools facilitate cross-border communication during international food safety events, why are they used, by whom, and for what purpose?

In this review, the proximal outcome of interest is the use of different tools to communicate internationally about issues related to food safety in an efficient manner. The distal outcomes of interest can be understood as the outcomes or consequences of using the tools. Some examples may include the identification of the source of an outbreak, facilitation of risk management actions in different countries and prevention of foodborne disease. Although important, it is beyond the scope of this review to examine and measure the impact that using different tools has on the overall safety of the global food supply. However, several insights relating to the utility of different tools to prevent or mitigate the burden of foodborne disease will be garnered from this review to be further explored in future research. The terminology used in the review is outlined in online supplementary file 2.

### Research aim and objectives

The primary aim of this synthesis is to address the question How do different tools facilitate cross-border communication during international food safety incidents, why are they used, by whom, and for what purpose? The overall objective is to refine a programme theory that explains the contexts (C) in which certain mechanisms (M) generate certain outcomes (O) by developing a series of C–M–O statements. This programme theory should prove useful

to programme coordinators to promote and support the use of communication tools and improve their effectiveness. The specific objectives are as follows:

1. Document the different tools used to facilitate cross-border communication during international food safety incidents.
2. Examine the outcomes observed in relation to the use of different communication tools.
3. Identify and explain the mechanisms that influence the outcomes observed in relation to the use of different communication tools.
4. Identify the contextual factors that trigger mechanisms to influence the outcomes observed in relation to the use of different communication tools.
5. Refine a realist programme theory that synthesises review findings and input from an expert reference committee to explain how different tools facilitate cross-border communication during international food safety events, why they are used, by whom and for what purpose?

### The realist approach

A realist approach has been chosen to conduct this review as it is well suited for the examination of complex programmes through its focus on outcomes in real-world settings and the contextual factors that influence them.[30] A realist perspective of social change underpins this approach whereby the actions of individuals and their understanding of the world serve to construct social phenomena and are influenced by cultural, institutional and social structures.[31 32] This interpretative method is theoretically driven and allows evidence from a range of study designs to be synthesised. The use of theory facilitates a deeper understanding with respect to policy intentions and appreciates the complexity of programmes by including context in the analysis.[33] The overall intent of a realist review is the development and refinement of programme theories to understand how context influences mechanisms to generate outcomes. Mechanisms can be understood as the underlying context-dependent processes, behaviours, structures, values or levers that are able to generate outcomes. The context includes the social, cultural, institutional, historical and environmental factors that form the setting in which actions are taken to trigger mechanisms. The resulting outcomes of the programme, system or intervention under examination are the products of certain mechanisms being triggered in certain contexts and may be intended or unintended.[30 34 35]

In this review, identifying mechanisms will help to explain how competent authorities use existing communication tools during international food safety events to exchange information across national borders. Taking the realist perspective, several C–M–O configurations may be articulated within the programme theory to explain this phenomenon. The C–M–O configurations will allow the research to be abstracted and applied to multiple contexts, bolstering external validity. The process of

theory building and configuring the C–M–O will be iterative, enabling the confirmation, refutation or modification of the initial programme theory.[30 33 36]

### Preliminary work to identify initial programme theory

To identify an initial programme theory, a range of sources have been used including the author's experiences as the current secretariat of the FAO/WHO INFOSAN, a scoping review of published papers describing international food safety events and grey literature pertaining to various food safety communication tools currently in use and elicitation of input from an international expert reference committee consisting of nine members including some coordinators of international communication tools currently in use (see the Acknowledgements section for details). This preliminary work has proposed an initial programme theory to suggest that when the context is such that a country: (1) is an importer or exported of food commodities; (2) has the technical infrastructure to detect food safety events (including foodborne disease outbreaks or food contamination) and (3) is governed in accordance with regional and/or global laws and regulations relating to food control and global health security, then certain mechanisms including trust, experience, support, awareness, understanding and a sense of community will facilitate the proximal outcome of using communication tools to relay information abroad and a potential range of distal outcomes, including: (1) inter-sectoral collaboration among different national stakeholders from agriculture, food and health authorities; (2) efficient exchange of information between international stakeholders; (3) timely detection, notification and response to food safety events (including the implementation of risk management measures); (4) reduction of food safety risks; (5) robust understanding of the international dimensions of a given food safety event and (6) prevention of foodborne disease. It is proposed that variations in the context will influence whether the proposed mechanisms will trigger the outcomes. A schematic overview of this initial programme theory is provided in figure 1.

## METHODS/DESIGN

This synthesis will adhere to the 2005 protocol provided by Pawson et al, for conducting realist reviews and reporting will be guided by the Realist and Meta-narrative Evidence Synthesis: Evolving Standards (RAMESES) from Wong et al.[33] The five steps for conducting a realist review according to Pawson et al[30] are as follows: (1) clarify scope; (2) search for evidence; (3) appraise primary studies and extract data; (4) analyse and synthesise evidence; and (5) disseminate. While presented sequentially, these steps are iterative and will be revisited throughout the review process when new evidence emerges that can contribute to theory refinement. The grand level development theories that provide an overarching framework for this review include the third wave of modernisation theory developed in the 1990s[37 38] and

globalisation theory as articulated by Robinson.[39] Both theories provide a lens through which to understand that though the world is becoming ever more interconnected and interdependent, certain structures built to support development cannot be imposed in exactly the same way at the same time in different countries because the country-specific context will influence the outcomes. Modernisation theory also helps to explain the development of systems and tools within societies. This is particularly relevant in the context of ensuring food safety as there are international food safety standards and guidelines (including guidelines for communication during international food safety events) that must be adopted in national settings to improve food safety systems and facilitate food trade. Globalisation theory helps to explain that with the introduction of international food safety standards and guidelines, national governments cannot operate in isolation if they wish to engage in food trade. With this understanding and using the realist approach, a refined programme theory will be developed to explain C–M–O configurations related to the use of communication tools to facilitate information exchange during international food safety events. Two reviewers will undertake this work and the expert reference committee will provide feedback during the review. The review will be conducted over a 12-month period from January 2019 to December 2019 (see figure 2 for an overview of the stages of this review).

### Search strategy

To test the initial programme theory, a systematic search of the literature will aim to identify documents written in English, dating back to 1995 that illuminate how different tools facilitate cross-border communication during international food safety events, why they are used, by whom and for what purpose. This search will be undertaken using the databases Web of Science, Embase, MEDLINE, PubMed and CINAHL. A comprehensive search algorithm has been developed with assistance from a librarian at Lancaster University, United Kingdom, by first selecting key search terms following the review of titles and abstracts from 10 known publications describing international food safety events or an international food safety communication tool, system or network. Combinations of the following key words in English (and their truncations where required) using Boolean operators and proximity operators (where possible) will be entered into the selected databases: (systems OR network OR tool OR communication OR notification OR "information exchange") AND (international OR multi-state OR multi-country OR imported OR exported) AND (("food safety" OR "food contamination" OR "foodborne diseases") OR (gastroenteritis AND (incident OR emergency OR outbreak)) OR (food AND (incident OR emergency OR outbreak))). See online supplementary file 3 for the specific database searches.

**Figure 1** To identify an initial programme theory, a range of sources have been used including the author's experiences as the current secretariat of the FAO/WHO INFOSAN, a scoping review of published papers describing international food safety events and grey literature pertaining to various food safety communication tools currently in use and elicitation of input from an international expert reference committee including some coordinators of international communication tools currently in use. FAO, Food and Agriculture Organization; INFOSAN, International Food Safety Authorities Network.

Bibliographic references from documents selected for inclusion will be reviewed using the snowballing method to identify other potentially relevant documents. Since grey literature can be a pertinent source of information for realist reviews, annual reports, evaluation summaries, or policy documents published by international organisations or government agencies will also be searched for on respective websites.[30] The grey literature search will be purposeful and undertaken on the organisational websites related to those tools that have been already identified during the scoping review or through discussions with the expert reference committee, or that are later identified following the database searching. Members of the expert reference committee will also be asked to provide any grey literature pertaining to such tools they believe may be relevant. The search for evidence will be driven by the research objectives and will be iterative in practice to identify all relevant information sources to develop the programme theory. Searching will conclude when theoretical saturation is reached and sufficient evidence has been collected to confidently assert that the proposed theory is plausible.[33] The expert reference committee will contribute to this research by identifying additional articles and documents for consideration in the review and will provide feedback on the emerging programme theories as they are developed and refined. The search strategy will also be reviewed iteratively by this committee to ensure the scope of the search is appropriately designed to achieve the overall research aim

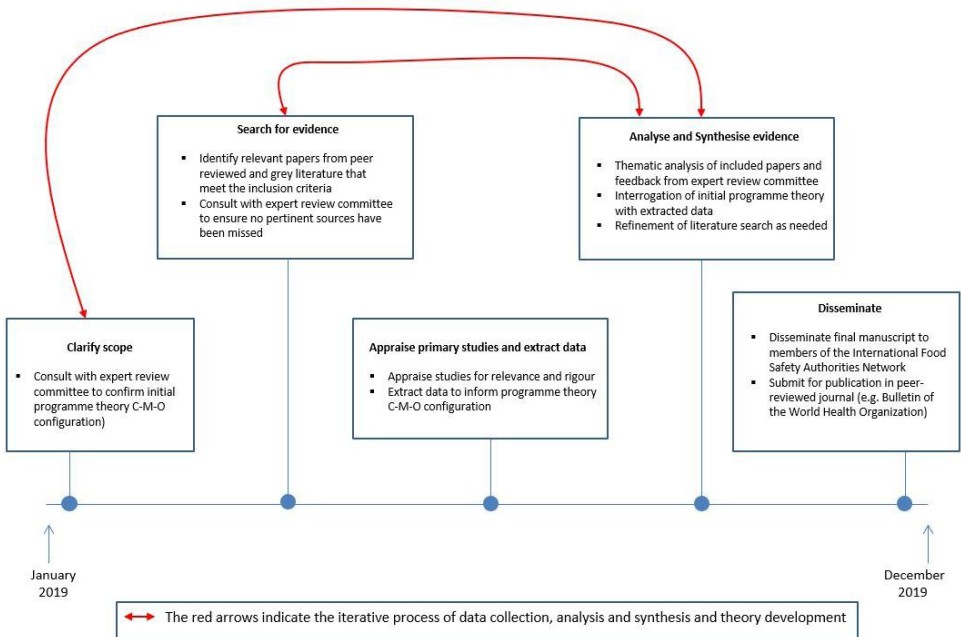

**Figure 2** The review will be conducted over a 12-month period from January 2019 to December 2019. C–M–O, context–mechanism–outcome.

and objectives. Throughout this process, references will be managed using Endnote X7 software.

### Study selection criteria and procedures

To ensure that the development of programme theory considers a wide range of evidence, it is customary to use broad inclusion/exclusion criteria in a realist synthesis.[30] The inclusion criteria are studies of any design from peer-reviewed literature and other documents from grey literature that are written in English, published in 1995 or later, describe an international food safety event or a communication tool and provide evidence that contributes to the synthesis and the emerging programme theory. The year 1995 was chosen because tools used prior to this are more likely to reference outdated technology (eg, facsimile) that would not be relevant in today's internet-dependent world. The exclusion criteria are if a document does not describe an international food safety event or a communication tool with sufficient details to inform the programme theory or focuses on an outdated communication technology (eg, facsimile). Two reviewers will independently screen the title and abstract of the searched studies using the inclusion and exclusion criteria to maintain rigour in this review. If it is unclear from the title and abstract if a paper should be included (or if the paper does not have an abstract as with many documents from grey literature), the full text will be reviewed prior to exclusion. Differences will be discussed by the two reviewers and disagreements will be resolved through discussion with the expert reference committee until consensus is reached. This process will facilitate dialogue among reviewers and the expert reference committee in an effort to include all relevant data.

### Data extraction and study appraisal

In realist synthesis, data extraction is more akin to note taking.[30] Each document included in the study will be reviewed using a bespoke data extraction form in Microsoft Excel to facilitate and organise note taking (online supplementary file 4). The variables extracted will include: (1) title; (2) authors; (3) year of publication; (4) type of document/study (5) countries involved; (6) international/regional organisations involved; (7) specific foodborne hazard; (8) implicated food item; (9) name and details of communication tool used; (10) contextual factors that facilitated the use of the tool; (11) contextual factors that limited the use of the tool; (12) conclusions made by the authors with respect to the use of the tools; (13) recommendations made by the authors with respect to improving international communication during international food safety events; (14) any other contextual factors; (15) any other underlying mechanisms and (16) points of discussion to raise with expert reference committee. The use of this form is intended to focus on the extraction of information about contexts, mechanisms and outcomes on that which specifically contributes to the refinement of the initial programme theory.

As per the RAMESES guidelines, the quality appraisal will be made on the basis of how each study contributes to the development of C–M–O configurations.[33] In a realist synthesis, quality is determined by assessing two criteria: (1) relevance and (2) rigour.[40] Relevance refers to the degree to which the information in the study fits within the scope of the review and rigour refers to methodological rigour and the degree to which conclusions reached in the study are appropriately drawn based on the research design employed.[30] To aid in this assessment, the

Mixed Methods Appraisal Tool (MMAT)[41] will be used, but will only be applied to the relevant aspects of each study under review and not necessary the whole study. This tool allows for assessment of multiple study designs concurrently, it has theoretical and content validity and it has also been tested for efficiency and reliability.[41 42] To assess relevance, each document will be scored as one of the following categories (adopted from Wozney et al[43] and Flynn et al[44]): (1) low/no contribution; (2) medium contribution or (3) high contribution. Evidence will also be assessed as either objective (empirical) or subjective (anecdotal). The relevance and rigour of each of the included studies will be evaluated by two reviewers who will document a summary of their assessment in tabular format for consideration during analysis. Differences will be discussed by the two reviewers and disagreements will be resolved through discussion with the expert reference committee until consensus is reached. Documents will not be excluded based on the MMAT score, nor will documents from which evidence is anecdotal, but collecting this information will provide insight into the rigour of existing research in this field.

### Data synthesis

With consideration for abductive and retroductive analysis,[45 46] documents will be examined for evidence that support, refute or refine the initial programme theory. Synthesis will involve analysing data that were absent from the initial programme theory (abduction) and moving between theory and observable data (retroduction), enabling the formation of new ideas beyond the initial programme theory. Taking this approach will utilise both inductive and deductive analytic processes to understand the C-M-O configurations. A thematic approach will be applied to record patterns in context, mechanisms and outcomes within each document reviewed and then across documents. These patterns will be compared with the original programme theory to determine if they support, expand or refute its configuration. As articulated in the RAMESES guidelines, the intention here will be to interrogate the C-M-O configurations and not to provide quantifiable summary data from the studies reviewed.[21] If the reviewed data do not fully explain the initial theory or if new theories emerge through this process, the literature search will be refocused in order to adequately synthesise a final programme theory with supporting thematic explanations.

### Validity

Using an iterative approach to understand how different tools facilitate cross-border communication during international food safety events, why they are used, by whom, and for what purpose, will allow researchers to revisit the C-M-O configurations throughout the process as data from the literature is collected.[22] This practice and the intentional inclusion of context in the analysis will improve external validity and the potential generalisability of mechanisms identified in the review.[21] Further,

the utilisation of an expert reference committee to elicit feedback, identify additional publications and review the programme theories as they are developed, serves to further bolster internal validity.

### Ethics and dissemination

Ethical approval is not required for this review as it does not involve primary research. However, it will be conducted according to the appropriate ethical standards of accuracy, utility, usefulness, accountability, feasibility and propriety.[32]

The RAMESES publication standards will be followed to report the findings of this review. On completion, the final manuscript will be shared with members of INFOSAN, which includes public health and food safety professionals from national government agencies in 188 countries. Further, it is the intent of the author to submit the review for publication in a leading peer-reviewed journal focusing on globalisation and health. The review will also be submitted as a chapter in the first author's PhD thesis to be submitted to Lancaster University.

### Patient and public involvement

Patients were not involved in the design of this study.

## DISCUSSION
### Significance

Increasingly, globalisation of our food supply necessitates international communication and coordination among food safety and public health professionals to prevent, detect and respond to foodborne disease outbreaks and instances of food contamination that affect more than one country. Rigorous research is needed to understand how the various tools used to facilitate communication are working and in what contexts. The knowledge gained from this study will provide valuable lessons on how different tools facilitate cross-border communication during international food safety events, why they are used, by whom, and for what purpose.

### Limitations

One limitation of this review is that it will only be conducted in English and therefore may introduce an element of language bias. In addition, the formulation of the C–M–O programme theory relies heavily on published literature and therefore may be subject to publication bias. Finally, review findings will be context-specific and therefore must be considered within the context of this research.

## CONCLUSION

Responding to international food safety events is complex, in part because of the globalised nature of our food supply and the involvement of numerous international stakeholders. In this paper, a protocol for conducting a realist synthesis on different tools to facilitate

cross-border communication during international food safety events has been presented which has important but understudied implications on global efforts to mitigate the burden of foodborne illness resulting from internationally distributed food. The programme theory to be developed will be useful to policy-makers and those coordinating the operation of communication tools currently in use, who may adapt components of the tools according to different contextual factors to promote, support and improve their use. By improving international coordination and communication during international food safety events, the global burden of foodborne disease can be mitigated.

**Acknowledgements** The authors wish to acknowledge the members of expert review committee for their input during the development of the initial programme theory and for reviewing earlier drafts of this manuscript. Members include: Mr Jan Baele (RASFF Coordinator, European Commission, Belgium), Mr Juan Ortuzar (International Affairs Coordinator, Chilean Agency for Food Safety and Quality, Chile), Dr Caroline Merten (Scientific Officer, European Food Safety Authority, Italy), Ms Jenny Bishop (Team Manager, Food Compliance Services Group, Operations Branch, Ministry for Primary Industries, New Zealand), Dr Jorgen Schlundt (Director, Nanyang Technological University Food Technology Center, Singapore), Dr Peter Ben Embarek (Scientist, INFOSAN Secretariat, Department of Food Safety and Zoonoses, WHO, Switzerland), Mr Adam Bradshaw (Technical Officer, INFOSAN Secretariat, Department of Food Safety and Zoonoses, WHO, Switzerland), Mr Raul Garcia (Consultant, INFOSAN Secretariat, Department of Food Safety and Zoonoses, WHO, Switzerland) and Dr Peter Gerner-Smidt (Chief, Enteric Diseases Laboratory Branch, Centers for Disease Control and Prevention, USA).

**Contributors** CJS conceived the original idea, designed the study, drafted the manuscript and approved the final document. CM drafted the manuscript and approved the final document. CJS is a staff member of the World Health Organization. The authors alone are responsible for the views expressed in this publication and they do not necessarily represent the views, decisions or policies of the WHO.

**Funding** This study is funded by the World Health Organization.

**Competing interests** None declared.

**Patient consent for publication** Not required.

**Provenance and peer review** Not commissioned; externally peer reviewed.

**ORCID iD**
Carmen Joseph Savelli http://orcid.org/0000-0001-5929-7249

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
