## [Reviewer comments · BMJ Open]

ARTICLE DETAILS

TITLE (PROVISIONAL)	Utilisation of tools to facilitate cross-border communication during international food safety events, 1995-2019: A realist synthesis protocol
AUTHORS	Savelli, Carmen; Mateus, Ceu

VERSION 1 – REVIEW

REVIEWER	Dr Annabelle Wilson Flinders University, Australia
REVIEW RETURNED	23-May-2019

GENERAL COMMENTS	Congratulations to the authors on a well written paper, about an important topic. The realist approach to the review is novel and will glean useful information about context and outcome in relation to food safety communication and coordination across countries. The authors may be interested in a paper recently published by our research group on a related issue: Wilson, A., Tonkin, E., Coveney, J., Meyer, S., McCullum, D., Calnan, M., Kelly, E., O'Reilly, S., McCarthy, M., McGloin, A. and Ward, P., 2019. A cross-country comparison of strategies used to build consumer trust in the food supply. Health Promotion International.
---

REVIEWER	Rachel Flynn University of Alberta, Canada
REVIEW RETURNED	27-May-2019

GENERAL COMMENTS	Thank you for the opportunity to review this realist synthesis protocol. International food safety is not my substantive area, I have primarily focused my comments on the methods using the RAMSES quality standards. Background: The background sections provide a clear understanding of the importance of international food safety and the issue with communication tools and international food safety. The authors clearly define what they mean by a communication tool, however line 53-58 on page 5 of 18 requires more expansion "evidence from practice suggests that such tools are only effective..." Please expand on what contexts, geographical areas and what types of communication tools. I would suggest a key terms table early in this paper that defines, context, mechanism, outcome, communication tools and international food safety events.
---

	Line 13, page 6 of 18: Please cite some of the publications here and expand on this. Methods: Line 30 pg 7: macro-level development theories - please define what is meant by this term, are these middle-range theories or grand level theories that the authors will draw upon to guide the program theory, synthesis? How do these theories provide an overarching framework? This needs to be clarified with more detail on how. Line 42 pg 7: Here the authors state that they will develop a middle-range program theory, how? The end result from the realist synthesis is a refined program theory. Please clarify and explain. Line 53 pg 7: I am concerned about the feasibility of the timeline proposed, is there a reason for 6 months? I would suggest that a year timeline is more feasible, if 6 months is a must, the authors should explain how they will manage the scope, search and papers retrieved to manage this tight timeline. Line 5 pg 8: The authors mention complex program - please explain how the communication tools being studied are complex - this should be addressed in the background section. Line 20 pg 8: Here the authors state the intent of a realist review is the development of program theories, yet they state they will develop a MRT, please explain. Line 36 pg 8: How will the authors identify what is a context, a mechanism or an outcome - it may be useful to provide some potential examples or the plan of how authors will make these judgement - suggest drawing on IPT already developed. Line 47 pg 8: Please explain the process of how the authors will generate C-M-O configurations, including any software to manage this. The realist approach section should come after the research objectives and aims and before methods details on the search etc Research aim and objectives: Would move this section to come before methods In this section the authors state that the purpose is to refine a program theory - use this consistently throughout the paper. Line 26 pg 10: Objective 5: should state: refine a program theory rather than develop Identifying initial program theory: This should be presented as preliminary work prior to discussing the methods section for the realist review. Suggest presenting the initial program theory in the form of CMO configurations or hypotheses. Search strategy: Justify the date limits Line 10, pg 12: Good detail on the database searching, limited detail on the grey literature searching, more detail is needed on what international organisations, government agencies and websites will be searched, what terms will be searched to identify the relevant grey literature, what is the inclusion/ exclusion criteria for the grey literature, how will the search and number of retrievals be managed, especially when there is a 6 month timeline to
--	--

	complete the review- these details are important to describe in more detail than currently provided. Line 18 pg 12: Please provide more detail on the expert reference panel - how many members, role and level of engagement in the review process. Line 5 pg 12: Should state development and refinement Line 41 pg 12: need to define international food safety event and what contexts. Also should all documents describe a communication tool, the authors state or a communication tool. Line 44 pg 12: Please describe how the authors will determine if a paper provides evidence that contributes to CMO Line 50: This describes the process suited for empirical or database searches, what process will the authors follow for grey literature- or items that do not have an abstract. Please describe how judgement will be made about including/ excluding data, what is the justification. Data extraction and appraisal: Line 12 pg 13: Remove sentence about note taking or link this sentence to the extraction approach. Line 14: Describe the bespoke extraction form- what software will be used and please describe extraction of C, M, O configurations, should 10 & 11 state contextual factors? what potential mechanisms and outcomes will the authors extract- please make explicit here. Line 38 pg 13: How will the authors identify, distinguish and present whether information in a document is of high relevance or other wise and same for high rigour or otherwise, this needs to be detailed. How will this information be taken into account during analysis and synthesis? The MMAT is useful for empirical studies to show level of methodological quality but this does not address level of CMO contribution in terms of realist relevance and rigour or for secondary sources. Relevance – whether it can contribute to theory building and/or testing; and Rigour – whether the method used to generate that particular piece of data is credible and trustworthy. Data synthesis: Explain the abductive process, more detail on retrodution in realist synthesis. Please describe in more detail how the team manage this synthesis process. More detail is needed to explain how the authors will carry out this process. Ethics: Unsure the relevance of this section. If required elaborate on the ethical standards, detail how the review will be conducted according to these standards. Limitations: This is not seen as a limitation in realist review as that is not its intention, instead think how the refined program theory can be further tested.
--	--

VERSION 1 – AUTHOR RESPONSE

Reviewer 1

Comment	Response
Congratulations to the authors on a well written paper, about an important topic. The realist approach to the review is novel and will glean useful information about context and outcome in relation to food safety communication and coordination across countries. The authors may be interested in a paper recently published by our research group on a related issue: Wilson, A., Tonkin, E., Coveney, J., Meyer, S., McCullum, D., Calnan, M., Kelly, E., O'Reilly, S., McCarthy, M., McGloin, A. and Ward, P., 2019. A cross-country comparison of strategies used to build consumer trust in the food supply. Health Promotion International.	Many thanks for taking the time to review this paper and for providing the article recently published by your research group. This is much appreciated.

Reviewer 2

Comment	Response
Thank you for the opportunity to review this realist synthesis protocol. International food safety is not my substantive area, I have primarily focused my comments on the methods using the RAMSES quality standards.	Many thanks for your thorough review and for all your helpful comments which have been addressed throughout the paper and are explained below. The authors wish to express their gratitude to you for taking the time to provide such detailed feedback.
Background: The background sections provide a clear understanding of the importance of international food safety and the issue with communication tools and international food safety. The authors clearly define what they mean by a communication tool, however line 53-58 on page 5 of 18 requires more expansion "evidence from practice suggests that such tools are only effective..." Please expand on what contexts, geographical areas and what types of communication tools.	We have added the following text to expand on what is meant by saying that certain tools operate in certain contexts and geographic areas: "Some examples of these communication tools include the European Rapid Alert System for Food and Feed (RASFF), the Association of Southeast Asian Nations (ASEAN) RASFF, and the International Food Safety Authorities Network (INFOSAN). The European RASFF system is an example of a regional tool that works in the European context, in part because member-countries adhere to the same legislation. The ASEAN RASFF system is an example of a tool that is less well established because member-countries in this Asian context do not adhere to the same legislation. INFOSAN is a global tool, coordinated by the World Health

	Organization (WHO) and the Food and Agriculture Organization of the United Nations (FAO), but as described by Savelli, Bradshaw, Ben Embarek, and Mateus²⁰, a relatively limited number of active members from a select group of countries contribute most information exchanged through the network. Supplementary file 1 provides a preliminary inventory of communication tools currently used or under development for exchanging information during international food safety events.” We believe that by giving these three examples, readers will be clear on the types of communication tools under consideration. In addition, supplementary file 1 provides additional details for those who wish to know more.
I would suggest a key terms table early in this paper that defines, context, mechanism, outcome, communication tools and international food safety events.	As suggested, these key terms have been added in a table in supplementary file 2 and the following text has been added: “The terminology used in the review is outlined in supplementary file 1”
Line 13, page 6 of 18: Please cite some of the publications here and expand on this.	As suggested, we expanded here to say, “These papers are typically written as outbreak reports rather than research studies” and have cited 10 publications describing international food safety events.
Line 30 pg 7: macro-level development theories - please define what is meant by this term, are these middle-range theories or grand level theories that the authors will draw upon to guide the program theory, synthesis? How do these theories provide an overarching framework? This needs to be clarified with more detail on how.	Macro-level has been removed and replaced with grand level, as that was what the original text was intended to mean – referring really to a substantive theory. As suggested, the following text has been added to expand on how these theories provide an overarching framework: “Modernisation theory also helps to explain the development of systems and tools within societies. This is particularly relevant in the context of ensuring food safety as there are international food safety standards and

	guidelines (including guidelines for communication during international food safety events) that must be adopted in national settings to improve food safety systems and facilitate food trade. Globalization theory helps to explain that with the introduction of international food safety standards and guidelines, national governments cannot operate in isolation if they wish to engage in food trade.”
Line 42 pg 7: Here the authors state that they will develop a middle-range program theory, how? The end result from the realist synthesis is a refined program theory. Please clarify and explain.	Thanks for pointing this out – middle-range has been replaced with refined program theory. The authors note that there are some differences in the way in which “program theory” has been conceptualized in the literature which led to our use of the original terminology. As explained by Shearn and colleagues (2017): These differences are due in part to the fact that such theories can either represent a highly specific causal explanation or a more abstract explanation. Pawson (2010, 2013), for example, uses program theory somewhat interchangeably with middle-range theory, which is at a higher level of abstraction and can be generalized across different contexts. Other scholars make a distinction between program theory and middle-range or grand theories, by which they mean abstract theories that are not attached to a specific context (Davidoff et al., 2015). In conclusion, the original intention was indeed a refined program theory, so the text in the paper has been revised to reflect this.
Line 53 pg 7: I am concerned about the feasibility of the timeline proposed, is there a reason for 6 months? I would suggest that a year timeline is more feasible, if 6 months is a must, the authors should explain how they will manage the scope, search and papers retrieved to manage this tight timeline.	The initial timeline of 6 months was indeed overly ambitious. The timeline of 1 year has been adopted as suggested by the reviewer and the text revised on page 7 to reflect this change: “...over a 12-month period from January 2019 to December 2019...” as well as figure 1 which provides an overview of the stages of this review.
Line 5 pg 8: The authors mention complex program - please explain how the communication tools being studied are complex	The following text has been added to explain why these tools are complex: “In addition, these tools are complex for several reasons, including

- this should be addressed in the background section.	because they represent disparate systems that may or may not interface with each other, operate in different languages, are coordinated by different institutions in different countries and are at various stages of development.”
Line 20 pg 8: Here the authors state the intent of a realist review is the development of program theories, yet they state they will develop a MRT, please explain.	Thank you for pointing this out. The discrepancy has been resolved in the text by only referring to programme theory. MRT is no longer mentioned (as per explanation provided above).
Line 36 pg 8: How will the authors identify what is a context, a mechanism or an outcome - it may be useful to provide some potential examples or the plan of how authors will make these judgements - suggest drawing on IPT already developed.	The authors believe this is already explained with the following text: “To identify an initial programme theory, a range of sources have been utilised including the author’s experiences as the current secretariat of the FAO/WHO International Food Safety Authorities Network (INFOSAN), a scoping review of published papers describing international food safety events and grey literature pertaining to various food safety communication tools currently in use and elicitation of input from an international expert reference committee consisting of 9 members including some coordinators of international communication tools currently in use (see acknowledgement section for details). “ And since Figure 2 already provides examples, we think this should be sufficient, given limitations in word count.
Line 47 pg 8: Please explain the process of how the authors will generate C-M-O configurations, including any software to manage this.	During this process, different CMO configurations will be documented when supporting evidence is identified. No specific software is going to be used apart from word processing software so this doesn’t seem relevant to add.
The realist approach section should come after the research objectives and aims and before methods details on the search etc	The realist approach section has been moved after the Research aim and objective section as suggested.
Research aim and objectives: Would move this section to come before methods.	The Research aim and objectives section has been moved before the methods section as suggested.

In this section the authors state that the purpose is to refine a program theory - use this consistently throughout the paper.	Thank you for pointing out this inconsistency – develop has been replaced with refine as suggested throughout the paper.
Line 26 pg 10: Objective 5: should state: refine a program theory rather than develop	Develop has been replaced with refine as suggested.
Identifying initial program theory: This should be presented as preliminary work prior to discussing the methods section for the realist review.	The title of this section has been revised as “Preliminary work to identify initial programme theory” and has been moved before the methods section as suggested.
Search strategy: Justify the date limits	The following text has been added to justify the date limit: The year 1995 was chosen because tools utilized prior to this are more likely to reference outdated technology (e.g. facsimile) that would not be relevant in today’s internet-dependent world.
Line 10, pg 12: Good detail on the database searching, limited detail on the grey literature searching, more detail is needed on what international organisations, government agencies and websites will be searched, what terms will be searched to identify the relevant grey literature, what is the inclusion/ exclusion criteria for the grey literature, how will the search and number of retrievals be managed,	The following text has been added to provide more details on the grey literature search: “The grey literature search will be purposeful and undertaken on the organizational websites related to those tools that have been already identified during the scoping review or through discussions with the expert reference committee, or that are later identified following the database searching. Members of the expert reference committee will also be asked to provide any grey literature pertaining to such tools they believe may be relevant.” Inclusion/exclusion criteria is the same for all documents.

especially when there is a 6 month timeline to complete the review- these details are important to describe in more detail than currently provided.	
Line 18 pg 12: Please provide more detail on the expert reference panel - how many members, role and level of engagement in the review process.	Because full details of the expert review committee are already included in the acknowledgement section, we have added the following text to the section "Preliminary work to identify initial programme theory" to make it easier to find this information: "...consisting of 9 members including some coordinators of international communication tools currently in use (see acknowledgement section for details)." The authors feel that the role of the expert reference committee and their level of engagement is already adequately described throughout the paper in the appropriate sections and would prefer not to add more details in this section due to limitations in word count.
Line 5 pg 12: Should state development and refinement	This has been amended accordingly to now read, "developed and refined".
Line 41 pg 12: need to define international food safety event and what contexts. Also should all documents describe a communication tool, the authors state or a communication tool.	International food safety events is now defined in the table of key terms as previously suggested. "Or" is correct here because some of the documents describing an international food safety event may provide evidence for the programme theory without explicitly mentioning a tool/network/system.
Line 44 pg 12: Please describe how the authors will determine if a paper provides evidence that contributes to CMO	This is now described later: "To assess relevance, each document will be scored as one of the following categories (adopted from Wozney et al and Flynn et al): 1) low/no contribution; 2) medium contribution; or 3) high contribution. Evidence will also be assessed as either objective (empirical) or subjective (anecdotal)."
Line 50: This describes the process suited for empirical or database searches, what process will the authors follow for grey literature- or	

items that do not have an abstract. Please describe how judgement will be made about including/ excluding data, what is the justification.	As per the explanation above, the search for grey literature will be limited and as such full documents from the grey literature will be reviewed before judgement is made about inclusion or exclusion and the same criteria applies for all documents and we replaced the word study with document to reflect this: “The exclusion criteria are if a study document does not describe an international food safety event or a communication tool with sufficient details to inform the programme theory or focuses on an outdated communication technology (e.g. facsimile).” Further clarified by adding the following italicized text: “If it is unclear from the title and abstract if a paper should be included (or if the paper does not have an abstract as with many documents from grey literature), the full text will be reviewed prior to exclusion.”
Data extraction and appraisal: Line 12 pg 13: Remove sentence about note taking or link this sentence to the extraction approach.	The authors would prefer to keep this sentence as it is linked to the following sentence and explains how the data extraction form will be used, and this is in line with the methods proposed by Pawson et al in reference 22 as cited here.
Line 14: Describe the bespoke extraction form- what software will be used and please describe extraction of C, M, O configurations, should 10 & 11 state contextual factors? what potential mechanisms and outcomes will the authors extract- please make explicit here.	The extraction form will be in “Microsoft Excel” and that text was added. We don’t believe further description is necessary because it’s also included as a supplementary file. Yes, “contextual” was added for 10 and 11. We have made it explicit what will be extracted by adding the following text: “The use of this form is intended to focus on the extraction of information about contexts, mechanisms and outcomes on that which specifically contributes to the refinement of the initial programme theory.”
Line 38 pg 13: How will the authors identify, distinguish and present whether information in a document is of high relevance or other wise and same for high rigour or otherwise, this needs to be detailed?	This has now been clarified with the addition of the following text: “To assess relevance, each document will be scored as one of the following categories (adopted from Wozney et al and Flynn et al): 1) low/no contribution; 2) medium contribution; or 3) high contribution. Evidence will also be assessed as either objective

How will this information be taken into account during analysis and synthesis? The MMAT is useful for empirical studies to show level of methodological quality but this does not address level of CMO contribution in terms of realist relevance and rigour or for secondary sources. Relevance – whether it can contribute to theory building and/or testing; and Rigour – whether the method used to generate that particular piece of data is credible and trustworthy.	(empirical) or subjective (anecdotal). The relevance and rigor of each of the included studies will be evaluated by two reviewers who will document a summary of their assessment in tabular format for consideration during analysis. Differences will be discussed by the two reviewers and disagreements will be resolved through discussion with the expert reference committee until consensus is reached. Documents will not be excluded based on the MMAT score, nor will documents from which evidence is anecdotal, but collecting this information will provide insight into the rigor of existing research in this field.
Data synthesis: Explain the abductive process, more detail on retroduction in realist synthesis. Please describe in more detail how the team manage this synthesis process. More detail is needed to explain how the authors will carry out this process.	The following text was added to elaborate on these concepts: “With consideration for abductive and retroductive analysis, documents will be examined for evidence that support, refute or refine the initial programme theory. Synthesis will involve moving between theory and data, analysing data that were absent from the initial programme theory (abduction) and moving between theory and data that can be observed (retroduction), enabling the formation of new ideas, beyond the initial programme theory. Taking this approach will utilize both inductive and deductive analytic processes to understand the C-M-O configurations.”
Ethics: Unsure the relevance of this section. If required elaborate on the ethical standards, detail how the review will be conducted according to these standards.	This section is indeed required by BMJ, though ethical approval for this work was not required. Due to limitations in word count, further elaboration here does not seem warranted and readers can refer to the referenced paper if they would like further detail on the ethical standards mentioned.
Limitations: This is not seen as a limitation in realist review as that is not its intention, instead think how the refined program theory can be further tested.	That limitation has now been removed. This section has been revised to reflect the limitations mentioned at the outset of the article. Revised text now reads, “One limitation of this review is that it will only be conducted in English and therefore may introduce an element of language bias. In addition, the formulation of the context-mechanism-outcome programme theory relies heavily on published literature and therefore may be subject to publication bias. Review findings will be context-specific and therefore must be considered within the context of this research.”